# The impact of strike action by Ghana registered nurses and midwives on the access to and utilization of healthcare services

Perpetual Ofori Ampofo[1], David Tenkorang-Twum[1,2], Samuel Adjorlolo[3,4]*, Margaretta Gloria Chandi[5], Francis Kwaku Wuni[6], Ernestina Asiedu[7], Vida Ami Kukula[8], Sampson Opoku[9]

1 Ghana Registered Nurses and Midwives' Association of Ghana, Head Office, Okponglo, Accra, Ghana, 2 Department of Adult Health, School of Nursing and Midwifery, University of Ghana, Legon, Accra, Ghana, 3 Department of Mental Health, School of Nursing and Midwifery, University of Ghana, Legon, Accra, Ghana, 4 Research and Grant Institute of Ghana, Accra, Ghana, 5 GA West Municipal Health Directorate, Accra, Ghana, 6 Bolgatanga Regional hospital, Bolgatanga, Upper West Region, Ghana, 7 Department of Child and Maternal Health, School of Nursing and Midwifery, University of Ghana, Accra, Ghana, 8 Dodowa Health Research Center, Dodowa, Accra, Ghana, 9 Ghana Registered Nurses and Midwives Association. Koforidua, Eastern Region, Ghana

* sadjorlolo@ug.edu.gh

## Abstract

### Background

As the largest professional group, nurses and midwives play instrumental roles in healthcare delivery, supporting the smooth administration and operation of the health system. Consequently, the withdrawal of nursing and midwifery services via strike action has direct and indirect detrimental effects on access to healthcare.

### Objective

The current study examined the impact of strike action by nurses and midwives with respect to access to and use of health services.

### Method

Data were collected retrospectively from a total of 181 health facilities from all the 16 administrative regions of Ghana, with the support of field officers. Because the strike lasted for 3 days, the data collection span three consecutive days before the strike, three days of the strike and three consecutive days after the strike. Data analysis was focused comparing the utilization of healthcare services before, during and after strike. Data were analysed and presented on the various healthcare services. This was done separately for the health facility type and the 16 administrative regions.

### Findings

The results showed that; (1) the average number of patients or clients who accessed healthcare services reduced drastically during the strike period, compared with before the strike.

**Data Availability Statement:** All relevant data are within the manuscript and its Supporting information files.

**Funding:** The authors received no specific funding for this work.

**Competing interests:** The authors have declared that no competing interests exist.

Majority of the regions recorded more than 70% decrease in service use during the strike period; (2) the average number of patients or clients who accessed healthcare services after the strike increased by more than 100% across majority of the regions.

## Conclusion

The study showed that strike action by nurses and midwives negatively affected access to and utilization of healthcare services.

## Introduction

The services provided by nurses and midwives are essential in the attainment of universal healthcare coverage; hence the withdrawal of such services via strike action has direct and indirect detrimental effects [1]. The current study examines the impact of strike action declared and undertaken by members of Ghana Registered Nurses and Midwives Association (GRNMA), the umbrella body of nurses and midwives in Ghana. Strike is the last weapon used by employees to fight for salary and improved working conditions. Employees who feel their rights and wellbeing have been violated, interfered with, or disregarded by their employer may decide to strike as the only practical way to get their complaints acknowledged and addressed [2]. Strike actions by healthcare workers take varying forms such as suspension of general healthcare delivery services, provision of only emergency services, cessation of work for few hours, days, weeks and months [3, 4].

The decision to embark on strike is influenced by a myriad of factors, predominant among them is conditions of service. For example, among Ghanaian nurses, dissatisfaction with working conditions, particularly salary, was cited as the motivating factor for their strike actions. More specifically, the nurses expressed concern about the large wage disparity in reference to the medical doctors, rather than equal pay [5]. The connection between remuneration of health workers and the emergence of strike actions is further supported by a study in Nigeria. It has been shown that, the majority of strikes in Nigeria are primarily motivated by a desire for increased pay [6]. In Tanzania, the request for improved working conditions in the areas of infrastructure, medications, equipment, and other medical supplies, as well as underpayment of salary and allowances were the reasons for doctors' strike in 2012 [7]. Strike actions by healthcare workers in Egypt in 2011 was intended to persuade authorities to reinforce safety in healthcare facilities, increase healthcare budget, and improve on the deteriorating state of facilities in the country's health care system [8].

Other factors associated with strike actions include: poor working environment, demotivation among staff, delayed or unfairness in promotions [5, 6, 9]. Dissatisfaction with leadership and management has also been implicated in strike action by healthcare workers in Nigeria [6]. Notable among the managerial inefficiencies include delayed promotions and limited educational and training support [10]. Other studies have registered solidarity with a colleague for an unlawful assault, or dismissal as reasons why health workers may embark on strike action [9, 10].

Restricted access to healthcare, occasioned by strike, is a recipe for increased mortality and morbidity, decreased revenue mobilization and other undesirable outcomes. In Nigeria, strike action by healthcare workers potentiated increased referrals to private health facilities (66.0%) and its resultant challenges such as substandard treatment and high cost of services [6]. In a related study in Kenya involving nurses, Njuguna [11] reported that the strike action had a

negative impact on vaccination services for children. For example, there was a 56.9% decrease in the number of newborns who received vaccinations during the strike in public health facilities. However, the proportion of immunizations in Faith–based hospitals increased by 252%. Scanlon, Maldonado [12] also established a robust link between strike action by healthcare workers and poor maternity and health care utilization, including ANC and delivery at a health facility among pregnant women, and delays in a child's first oral polio vaccination in Kenya. In a related Kenyan study, outpatient attendance declined by 64.4%, special clinics attendance by 74.2%, deliveries by 53.5% and inpatient admissions by 57.8% when nurses and other healthcare professionals embarked on strike [11]. A decrease in health services utilization and admissions during the period of strike by nurses and medical doctors in Kenya contributed to a decrease in mortality rates registered across hospitals in Kenya [11, 13]. Similar findings have been reported in high income countries, including US [14], England [15, 16] and Finland [17]. For example, Ruiz et al. (2013) in their study in England found that compared with the non-strike period, emergency admissions fell by 2.4% while the elective admissions decreased by 12.8%. The authors reported a 7.8% drop in the number of outpatients seen by medical staff on the day of the strike and a 45.5% increase in the number of cancelled appointments by NHS hospitals, while accident and emergency attendances dropped by 4.7% [16].

The GRNMA, like other professional associations and labor unions, has embarked on several strike actions since 1978 as a broader strategy to obtain improved and better conditions of services for her members [5]. The most recent strike action embarked on by the GRNMA and allied health professionals recently was from September 21st to 23rd, 2020 after negotiations with the Government (employer) for improved conditions of service yielded no results. As noted above, strike actions generally leads to a reduction in the number of people accessing healthcare services [10, 18]. However, our understanding of the extent of disruption in access to a range of healthcare services is limited as majority of previous studies have tended to focus their attention on services rendered by a unit or department in hospital or health facility. Ghana's healthcare delivery operates on three levels, thus primary, secondary, and tertiary. At the primary level, both preventive and promotive services are provided by Community health planning services (CHPS) centers, health centers, and polyclinics. They serve a population of approximately 20,000 people. At the secondary level, curative services are provided by district hospitals and serve an average population of 100,000 to 200,000 people whiles at the tertiary level, curative and rehabilitative services are delivered at the regional/teaching hospitals which serve approximately an average population of about 1.2 million. The healthcare delivery system has witnessed an increase in the number of healthcare professionals. For example, nurse-patient ratio has improved from 1:2,172 in 2013 to 1:505 in 2017 with the total number of nursing workforce increasing from 12,245 in 2013 to 58,608 in 2017. Again, the midwife to women in fertile age (WIFA) ratio has improved from 1:1533 to 1:704 with the total number of midwives increasing from 4,185 to 9884. As the largest workforce in the healthcare system, nurses and midwives have significant roles to contribute to the attainment of universal health coverage (UHC). Therefore, understanding how strike actions by these professional groups impact the range of healthcare services available to the public will equip stakeholders to take appropriate steps to respond appropriately and timely to strike notices, avert strike actions to safeguard the health of the population. Second, although nurses and midwives and other health professionals in Ghana do embark on strike, there is no comprehensive and systematically generated evidence on the impact of their strike actions on access to healthcare. The lack of empirical data in this regard makes it extremely difficult for the nurses and midwives association (i.e., GRNMA) to quantify the contributions of her members to healthcare delivery and to make a compelling case for better conditions of service. At the same time, the employer is unable to readily envisage the devastation to access to healthcare delivery occasioned by the strike action

of nurses and midwives and other health professionals. The current study fills this void by investigating the impact of the 2020 strike action by GRNMA and its Allied health professionals on access to a range of healthcare services in Ghana.

## Study objectives

The objectives of the study are as follows;

1. Determine the number of people accessing and utilizing health services before, during and after strike action by nurses and midwives.

2. Compare the number of people who accessed essential healthcare services before, during and after the strike was called off.

## Method

### Project setting

Data were gathered from health facilities located across the 16 regions in Ghana and across all the levels of healthcare. As noted previously, the healthcare system in Ghana is structured. At the highest level is the teaching hospitals, followed by regional hospitals, district hospital, polyclinic, health centers or clinic and CHPS. Data were collected from health facilities that are under the auspices of Ministry of Health and Ghana Health Service (Collectively referred to as Government facilities) and Christian Health Association of Ghana (collectively termed CHAG facilities). Private health facilities were excluded from the study since the working conditions of nurses at these facilities differ from the counterparts in Government or CHAG facilities. In this study, majority of the facilities were owned by government ($n$ = 155, 85.6%), whereas 26 (14.4% CHAG facilities.

### Recruiting and training field officers

A total of 112 field officers were recruited across the 16 regions in Ghana through the regional offices of the GRNMA. A dedicated WhatsApp platform was created for the field officers and the research team to facilitate communication relating to the project. A training workshop was organized and held via the Zoom videoconference platform at the convenience of the field officers who were working as nurses in their respective health facilities. The major area for the training was the data gathering process, including how to maintain data integrity, avoid data contamination as well as ensure ethically responsible research conduct. This was intended to ensure quality data gathering and transmission via a dedicated electronic portal powered by Google.

### Study variables

The study variables, operationalized as the services rendered by the health facilities, were decided by the research team members following a series of meetings and consultations with researchers, policy makers and practitioner. The team also took into consideration local health priorities and the demands of the Sustainable Development Goal (SDG) 3. The research team unanimously agreed on the following study variables; (1) outpatient department services, (2) admissions, (3) deliveries, (4) surgical services, (5) reproductive health services, and (6) antenatal clinic (ANC) services. We defined ANC services as healthcare services delivered to pregnant women. While these may include reproductive health services, we also note instances where reproductive health services are delivered to non-pregnant women. Therefore, in this

study, we focus on reproductive health services as services accessible to non-pregnant women. The inclusion of delivery services, for instance, was in accordance with indicator 3.1 of the SGD3 (reducing "global maternal mortality ratio to less than 70 per 100,000 live births") and indicator 3.2 (reducing neonatal mortality to at least as low 12 per 1,000 live births. . . . . . ."). Reproductive health service was also included in view of indicator 3.7 of the SDG3: "universal access to sexual and reproductive health care services, including family planning. . . . . .". Lastly, the focus on antenatal services reflect indicator 3.1 of the SGD3 which is to reduce "global maternal mortality ratio to less than 70 per 100,000 live births" and indicator 3.2 that is concerned about reducing neonatal mortality to at least as low 12 per 1,000 live births and under-5 mortality to at least as low as 25 per 1,000 live births."

## Timeframe for project data

Data were collected before, during and after the strike period to allow for comparative analyses and discussions across different data collection points, with reference to the strike period. Because the strike lasted for three consecutive days (21st to 23rd September, 2020; Monday to Wednesday), we collected data for the 3-day period to appreciate the scale of the impact of the strike action. Besides, the number of clients or patients who accessed healthcare services differ from day to day. This made it difficult to restrict the data collection to any of the days the strike action occurred. To obtain a baseline data against which to assess the impact of the strike, we collected data for the 3 consecutive days before the strike action (14th to 16th September 2020; Monday to Wednesday). Lastly, data were collected 3 consecutive days after the strike action was called off (28th to 30th September 2020; Monday to Wednesday) to estimate the use of health services following the suspension of the strike action. The data collection lasted for approximately 2 months, spanning 10th October to 3rd December 2020.

## Data collection procedure

Data collection was aided by the tool designed by the research team based on the pre-determined study variables discussed previously. Prior to the data collection, institutional permission was sought from the various health facilities with introductory or permission letter issued by the GRNMA national secretariat to the field officers. As stated previously, data were collected for three consecutive days for each data collection period (e.g., before, during and after strike). The field officers completed the hardcopy of the questionnaire for each facility. Thereafter, they were provided with a dedicated link, powered by Google Form, where they inputted and transmitted the data electronically to a centralized receiver accessible to the research team. The same questionnaire was used across the health facilities. The field officers were informed to input nil or zero where the data sought for does not exist. For example, because CHPS compounds do not conduct surgeries, data on this service will not be available. The electronic form requires that the field officers provide additional information on the region, district, type of facility (e.g., hospital or health center) and ownership of facilities (e.g., Government or CHAG) where data were collected. Regular updates were provided on the WhatsApp page to keep the field officers informed about the submissions received.

## Ethical consideration

This is a retrospective study in which data on the number of people who utilized various healthcare services before, during and after strike action by nurses and midwives were gathered. The study did not involve direct human subject engagement. Rather, data were obtained from institutional archives as an aggregate data. The focus was on how many people visited or utilized healthcare services within the time frame above, without focusing on the

background or demographics of users of healthcare services. The data collected were also devoid of identifying information relating to the facilities. This means that neither the facility nor clients/patients will be identified. Thus, data were fully anonymized. The data on number of people accessing healthcare services is notably a public data in Ghana. The project was underpinned by other relevant ethical considerations in research, including confidentiality, data safety and data protection. Access to data was restricted to the research team or other individuals supporting the project, mainly data analysts. These individuals signed a statement confirming that they would adhere to the study procedures regarding confidentiality. The data collected were analyzed as regional aggregate data to further delink the healthcare facilities. The Institutional Review Committee of the Research and Grant Institute of Ghana has declared that given the nature and type of data collected, ethics approval prior to data collection was not necessary.

## Data management

By the end of the data collection process, a total of 191 submissions were received. However, some of the submissions were duplicates, perhaps because of the technical and internet connectivity issues. It was also observed that, some field officers did not provide the exact or absolute number of service users for the study variable. Instead, they provided inscription such as "over 400", making it difficult to determine the exact number of service users under reference. The dataset was subsequently cleaned by deleting the data anomalies or deviations, leaving a total of 181 submissions for analyses.

## Data analysis

The analysis of data proceeded on two key assumptions; (1) health facilities under the various categorization (e.g., hospital, health centers) in a region will be similar with respect to the average number of service users than those outside the region. That is, hospitals in Ashanti region will be similar in terms of the number of service users than hospitals in Volta region. This assumption is centered heavily on the variations in the population distribution across the regions which in turn influence the number of health service consumers; and (2) health facilities falling under a particular category will be similar in terms of the range of the services provided. For example, it was assumed that CHPS compound across the country will offer virtually the same type of health services. In the same vein, hospitals across the country are more likely to render the same set of health services. Any difference should be subtle or negligible. Based on the foregoing, data was analyzed at the health facilities level, segregated by region. That is, hospital data were analyzed on regional basis as were data from health centers.

To proceed, we computed the average number of service users for each region, taking into consideration the type of health facilities. For example, data from the hospitals in Ashanti region were summed and divided by the total number of hospitals that provided data for the study. This resulted in the average number of health service users from hospitals in Ashanti region. In instances where data were available for only one type of health facility in a region, the same data were used since the mean could not be calculated. Although the mean is sensitive to outliers, it is the most widely used descriptive statistics in research and publication. To address problems relating to outliers, we aggregated and analyzed data along regional framework and by nature of health facilities. Data was prepared using the IBM SPSS Version 23 and analyzed using excel. The analyses involve mostly descriptive statistics. We computed the percentage change in the average number of individuals accessing health services before, during and after strike.

## Results

### Distribution of health facilities

As noted previously, data were collected from a total of 181 health facilities. More than half of the facilities were hospitals ($n$ = 93; 51.4%), 64 were health centers (35.4%), 16 were CHPS compounds (8.8%) and 8 were polyclinics (4.4%). Table 1 showed the regional breakdown of

**Table 1. Number and type of health facilities across the regions.**

| Regions | Frequency | % | Regions | Frequency | % |
|---|---|---|---|---|---|
| **Ashanti** | | | **Greater Accra** | | |
| Hospital | 12 | 48 | Hospital | 6 | 54.5 |
| Polyclinic | 1 | 4 | Polyclinic | 2 | 18.2 |
| Health Center | 12 | 48 | Health Center | 2 | 18.2 |
| CHPS | 0 | 0 | CHPS | 1 | 9.1 |
| **Total** | **25** | **100** | **Total** | **11** | **100** |
| **Eastern** | | | **Ahafo** | | |
| Hospital | 5 | 100 | Hospital | 1 | 100 |
| **Total** | **5** | **100** | **Total** | **1** | **100** |
| **Western** | | | **Western North** | | |
| Hospital | 8 | 53.3 | Hospital | 4 | 36.4 |
| Polyclinic | 0 | 0 | Polyclinic | 1 | 9.1 |
| Health Center | 3 | 20 | Health Center | 5 | 45.5 |
| CHPS | 4 | 26.7 | CHPS | 1 | 9.1 |
| **Total** | **15** | **100** | **Total** | **11** | **100** |
| **Oti** | | | **Bono East** | | |
| Hospital | 4 | 100 | Hospital | 3 | 100 |
| **Total** | **4** | **100** | **Total** | **3** | **100** |
| **Volta** | | | **Bono** | | |
| Hospital | 9 | 81.8 | Hospital | 6 | 66.7 |
| Polyclinic | 0 | 0 | Polyclinic | 1 | 11.1 |
| Health Center | 2 | 18.2 | Health Center | 2 | 22.2 |
| **Total** | **11** | **100** | **Total** | **9** | **100** |
| **Northern** | | | **Savannah** | | |
| Hospital | 7 | 87.5 | Hospital | 2 | 18.2 |
| Health Center | 1 | 12.5 | Health Center | 6 | 54.5 |
| CHPS | 0 | | CHPS | 3 | 27.3 |
| **Total** | **8** | **100** | **Total** | **11** | **100** |
| **Upper West** | | | **Upper East** | | |
| Hospital | 9 | 37.5 | Hospital | 8 | 66.7 |
| Polyclinic | 1 | 4.2 | Polyclinic | 0 | 0 |
| Health Center | 8 | 33.3 | Health Center | 4 | 33.3 |
| CHPS | 6 | 25.0 | CHPS | 0 | 0 |
| **Total** | **24** | **100** | **Total** | **12** | **100** |
| **Central** | | | **North East** | | |
| Hospital | 7 | 31.8 | Hospital | 2 | 22.2 |
| Polyclinic | 1 | 4.5 | Polyclinic | 1 | 11.1 |
| Health Center | 13 | 59.1 | Health Center | 6 | 66.7 |
| CHPS Compound | 1 | 4.5 | CHPS | 0 | 0 |
| **Total** | **22** | **100** | **Total** | **9** | **100** |

the number of health facilities included in the project. With respect to the 93 hospitals, majority were in Ashanti region (*n* = 12), followed by Volta and Upper West (*n* = 9 each) and Upper East and Western (*n* = 8 each), with only in Ahafo region. Of the 64 health centers, majority were in Ashanti region (*n* = 12), followed by Central region (*n* = 13) and Upper West (*n* = 8), with only from Northern region. Of the 16 CHPS compounds, Upper West region contributed to six, followed by Western region (*n* = 4). There was no data from CHPS compounds located in several regions, including Ashanti, Oti, Bono, Eastern and Ahafo. Data were obtained from two polyclinics in Greater Accra and one each in the following regions: Ashanti, Western North, Bono, Upper West, North East and Central.

## Access to health services at hospitals before, during and after strike

### Outpatient department attendance

As shown in Table 2, the average number of patients accessing healthcare services at the outpatient department (OPD) reduced significantly during the strike, compared with before the strike. Taking Ahafo region as an example, an average of 363 patients attended the OPD. This number reduced drastically to 41 during the strike period. The percentage decrease in the mean OPD attendance for the regions ranges from 45.12% (Oti region) to 97.95% (North East region), whereas the percentage increase in mean OPD attendance after the strike was called off ranges from 83.40% (Bono region) to +100% (e.g., Ashanti region).

### Admissions

From the data presented in Table 3, the strike impacted negatively on hospital admissions. The average number of patients on admissions before and after the strike was more than twice the average number during the strike period. The percentage decrease in the mean admissions ranges from 46.42% (Bono region) to 94.58% (Upper West region). All the regions registered

**Table 2. Percentage change in the mean OPD attendance during and after strike.**

| Region | Before Strike | During Strike | | After Strike | |
|---|---|---|---|---|---|
| | Mean OPD Attendance | Mean OPD Attendance | % Δ from Before Strike | Mean OPD Attendance | % Δ from During Strike |
| Ahafo | 363 | 41 | -88.70 | 441 | +100 |
| Ashanti | 341.67 | 163.92 | -52.02 | 339.67 | +100 |
| Bono | 551.83 | 268.17 | -51.40 | 491.83 | 83.40 |
| Bono East | 702 | 383 | -45.44 | 712.67 | 86.08 |
| Central | 370.14 | 116.86 | -68.43 | 404.14 | +100 |
| Eastern | 426.8 | 159.6 | -62.61 | 477 | +100 |
| Greater Accra | 425.33 | 81 | -80.96 | 464.17 | +100 |
| North East | 317.5 | 6.5 | -97.95 | 341.5 | +100 |
| Northern | 475 | 26.57 | -94.40 | 660.14 | +100 |
| Oti | 333.5 | 183 | -45.12 | 361.5 | 97.54 |
| Savannah | 261 | 87 | -66.67 | 314.5 | +100 |
| Upper East | 293.38 | 100.63 | -65.70 | 311.13 | +100 |
| Upper West | 157.89 | 16.11 | -89.80 | 171.56 | +100 |
| Volta | 440.78 | 167.67 | -61.96 | 480.22 | +100 |
| Western | 397.37 | 183.13 | -53.91 | 435.32 | +100 |
| Western North | 130.75 | 56.25 | -56.98 | 154 | +100 |

**Table 3. Percentage change in the mean admissions during and after strike.**

| Region | Before Strike | During Strike | | After Strike | |
|---|---|---|---|---|---|
| | Mean Admission | Mean Admission | % Δ from Before Strike | Mean Admission | % Δ from During Strike |
| Ahafo | 71 | 5 | -92.96 | 73 | +100 |
| Ashanti | 57.5 | 26 | -54.78 | 46.67 | +79.5 |
| Bono | 116.33 | 62.33 | -46.42 | 110 | +76.48 |
| Bono East | 109.67 | 30.67 | -72.03 | 101.67 | +100 |
| Central | 81.71 | 36.86 | -54.89 | 65.29 | +77.13 |
| Eastern | 52.2 | 11.2 | -78.54 | 54.8 | +100 |
| Greater Accra | 50.17 | 11.83 | -76.42 | 52.67 | +100 |
| North East | 167.5 | 4 | -97.61 | 189 | +100 |
| Northern | 86 | 6 | -93.02 | 129.14 | +100 |
| Oti | 88.75 | 35 | -60.56 | 77.25 | +100 |
| Savannah | 42 | 20.5 | -51.19 | 54.5 | +100 |
| Upper East | 89 | 14.88 | -83.28 | 77.88 | +100 |
| Upper West | 75.78 | 4.11 | -94.58 | 67.33 | +100 |
| Volta | 120.33 | 32.11 | -73.31 | 105.33 | +100 |
| Western | 117.5 | 23.63 | -79.89 | 119.38 | +100 |
| Western North | 62.5 | 15.25 | -75.60 | 66 | +100 |

an increase in admissions after the strike was called off, with a percentage increase ranging 77.13% (Central region) to +100% (e.g., Greater Accra).

## Deliveries conducted

Table 4 showed that during the strike period, the average number of deliveries conducted across the regions was very low compared to the average number of deliveries conducted prior to the strike and after the strike. In Northern region, for example, an average of 25

**Table 4. Percentage change in the mean deliveries conducted during and after strike.**

| Region | Before Strike | During Strike | | After Strike | |
|---|---|---|---|---|---|
| | Mean Delivery | Mean Delivery | % Δ from Before Strike | Mean Delivery | % Δ from During Strike |
| Ahafo | 18 | 3 | -83.33 | 25 | +100 |
| Ashanti | 12.33 | 1.45 | -88.24 | 10.67 | +100 |
| Bono | 17.83 | 6.17 | -65.40 | 16.33 | +100 |
| Bono East | 11 | 0.67 | -93.91 | 12.67 | +100 |
| Central | 12.43 | 3.43 | -72.40 | 10.29 | +100 |
| Eastern | 14.4 | 3.4 | -76.39 | 12.8 | +100 |
| Greater Accra | 22.17 | 8.83 | -60.17 | 16.83 | 90.60 |
| North East | 19.5 | 4.5 | -76.92 | 20 | +100 |
| Northern | 25.71 | 2 | -92.22 | 40 | +100 |
| Oti | 9.5 | 6.25 | -34.21 | 14 | +100 |
| Savannah | 5 | 4.5 | -10 | 12 | +100 |
| Upper East | 38.5 | 5.38 | -86.03 | 17 | +100 |
| Upper West | 16.44 | 5 | -69.59 | 17 | +100 |
| Volta | 12.33 | 3 | -75.67 | 11.22 | +100 |
| Western | 12.13 | 5.63 | -53.59 | 13 | +100 |
| Western North | 7.75 | 1.25 | -83.87 | 9 | +100 |

deliveries were conducted prior to the strike. This reduced to 2 during the strike period but increased to 40 after the strike was called off. In terms of percentage change in deliveries conducted, Savannah region reported the lowest decrease (10%), whereas Bono East registered the highest decrease (93.61%). Apart from Greater Accra region, all the other regions recorded more than 100% increase in mean deliveries after the strike was called relative to the strike period.

## Surgeries performed

The data presented in Table 5 revealed that, the average number of surgeries performed decreased across the regions during the strike period, ranging from 67.27% (Savannah region) to 100% (e.g., North East). As can be seen in Table 5, no surgeries were performed in Ahafo and North East regions during the strike period. Whereas there was an increase in surgeries performed in most regions when the strike was called off relative to the strike period ($\geq$ 100% percentage increase in mean), only Savannah region recorded a decrease of 55.56%.

## Reproductive health services

Across the regions, it was observed that the average number of people who accessed reproductive health services dwindled dramatically during the strike period, compared with the periods before and after strike (Table 6). In three of the regions (i.e., Ahafo, Central and Northern), no one accessed reproductive health services during the strike period. The decrease in mean reproductive service use during the strike period, relative to before the strike ranges from 45.74% (Bono region) to 99.80% (Volta region). In contrast, the regions experienced a significant increase (i.e., $\geq$ 60.74%) in reproductive service use following the suspension of the strike, except in Savannah region where there was 96% decrease.

**Table 5. Percentage change in the mean surgeries performed during and after strike.**

| Region | Before Strike | During Strike | | After Strike | |
|---|---|---|---|---|---|
| | Mean Surgeries | Mean Surgeries | % Δ from Before Strike | Mean Surgeries | % Δ from During Strike |
| Ahafo | 1 | 0 | -100 | 2 | 100 |
| Ashanti | 6.25 | 1.75 | -72 | 5.42 | +100 |
| Bono | 18.33 | 4.17 | -77.25 | 19.17 | +100 |
| Bono East | 17.33 | 2.33 | -86.56 | 16 | +100 |
| Central | 7.14 | 1.29 | -81.93 | 6.57 | +100 |
| Eastern | 18 | 1.4 | -92.22 | 23.2 | +100 |
| Greater Accra | 12.67 | 3.33 | -73.71 | 42.5 | +100 |
| North East | 5 | 0 | -100 | 2.5 | 100 |
| Northern | 8.57 | 0.86 | -89.97 | 12.43 | +100 |
| Oti | 5.25 | 1.5 | -71.43 | 4.75 | +100 |
| Savannah | 27.5 | 9 | -67.27 | 4 | -55.56 |
| Upper East | 7.25 | 0.5 | -93.10 | 12 | +100 |
| Upper West | 6.22 | 1.22 | -80.39 | 6.33 | +100 |
| Volta | 8.33 | 0.89 | -89.32 | 8.56 | +100 |
| Western | 10.75 | 2.75 | -74.42 | 10.25 | +100 |
| Western North | 3.75 | 0.75 | -80 | 5.5 | +100 |

**Table 6. Percentage change in the mean reproductive service use (RSU) during and after strike.**

| Region | Before Strike | During Strike | | After Strike | |
|---|---|---|---|---|---|
| | Mean RSU | Mean RSU | % Δ from Before Strike | Mean RSU | % Δ from During Strike |
| Ahafo | 68 | 0 | -100 | 230 | 100 |
| Ashanti | 88.5 | 8.92 | -89.92 | 119.5 | +100 |
| Bono | 43 | 23.33 | -45.74 | 37.5 | 60.74 |
| Bono East | 63.67 | 5 | -92.15 | 48.67 | +100 |
| Central | 66.5 | 0 | -100 | 55.86 | 100 |
| Eastern | 21.6 | 0.4 | -98.15 | 59.8 | +100 |
| Greater Accra | 78.83 | 13.5 | -82.87 | 76.17 | +100 |
| North East | 107 | 0 | -100 | 161.5 | 100 |
| Northern | 38.33 | 0 | -100 | 60.33 | 100 |
| Oti | 27 | 1 | -96.30 | 37.75 | +100 |
| Savannah | 14 | 25 | 78.57 | 1 | -96 |
| Upper East | 51.75 | 6 | -88.41 | 55.38 | +100 |
| Upper West | 33.56 | 1.11 | -96.69 | 31.89 | +100 |
| Volta | 77.78 | 0.11 | -99.86 | 66.56 | +100 |
| Western | 58.25 | 2.25 | -96.14 | 65 | +100 |
| Western North | 59.5 | 16.25 | -72.69 | 53.25 | +100 |

## Antenatal clinic services

The average number of people who accessed antenatal clinic (ANC) services decreased greatly during the period of strike, compared with the period before and after the strike (Table 7). Ashanti region recorded the least percentage decrease in mean antenatal use (76.13%), whereas North East and Northern regions registered the highest decrease (100%). Relatedly, across the regions, there was more than 100% increase in antenatal service use following the suspension of the strike action, relative to during the strike.

**Table 7. Percentage change in the mean antenatal clinic service use (ANC) during and after strike.**

| Region | Before Strike | During Strike | | After Strike | |
|---|---|---|---|---|---|
| | Mean ASU | Mean ASU | % Δ from Before Strike | Mean ASU | % Δ from During Strike |
| Ahafo | 135 | 9 | -93.33 | 175 | +100 |
| Ashanti | 68.42 | 16.33 | -76.13 | 79.67 | +100 |
| Bono | 89 | 13.17 | -85.20 | 115.17 | +100 |
| Bono East | 215 | 67.67 | -68.53 | 234.33 | +100 |
| Central | 58.43 | 6.57 | -88.76 | 70.43 | +100 |
| Eastern | 63.2 | 2.2 | -96.52 | 124.6 | +100 |
| Greater Accra | 87.83 | 19.67 | -77.60 | 98 | +100 |
| North East | 150 | 0 | -100 | 187 | +100 |
| Northern | 164 | 0 | -100 | 291.71 | +100 |
| Oti | 60 | 16.75 | -72.08 | 70.75 | +100 |
| Savannah | 41.5 | 4.5 | -89.16 | 37.5 | +100 |
| Upper East | 27 | 0.5 | -98.15 | 30.25 | +100 |
| Upper West | 29.89 | 0.78 | -97.39 | 39.22 | +100 |
| Volta | 85.33 | 0.75 | -99.12 | 105.67 | +100 |
| Western | 104.5 | 24.88 | -76.19 | 117.5 | +100 |
| Western North | 34.5 | 1 | -97.10 | 48 | +100 |

**Table 8. Percentage change in outpatient department attendance during and after strike.**

| Region | Before Strike | During Strike | | After Strike | |
|---|---|---|---|---|---|
| | Mean OPD Attendance | Mean OPD Attendance | % Δ from Before Strike | Mean OPD Attendance | % Δ from During Strike |
| Ashanti | 299 | 55 | -81.61 | 409 | +100 |
| Bono | 102 | 0 | -100 | 110 | +100 |
| Central | 128 | 98 | -23.44 | 133 | +100 |
| Greater Accra* | 66 | 7.5 | -88.64 | 38.5 | +100 |
| North East | 142 | 0 | -100 | 144 | +100 |
| Upper West | 65 | 8 | -87.69 | 62 | +100 |
| Western North | 17 | 24 | 41.18 | 14 | +100 |

* = mean/average data

## Access to health services at polyclinics during strike

### Outpatient department attendance

The data on OPD attendance in Table 8 revealed that the average number of people accessing OPD services reduced drastically during the strike period, compared with before strike. The percentage decrease in OPD attendance ranges from 23.44% (Central region) to 100% in Bono region. Taking Ashanti region as example, before the strike, 299 people accessed OPD services. However, this dropped to 55 during the strike period, representing 81.61% decrease. The results further showed that, OPD attendance increased above 100% across the regions when the strike was called off.

### Admissions

As shown in Table 9, the average number of patients on admissions declined during the strike period across the regions. For example, In Bono, Greater Accra and North East regions, there were zero admissions during the strike period. All the regions witnessed an increase in admissions after the strike, ranging from 41.67% to +100%.

### Deliveries conducted

The average number of deliveries conducted reduced during the strike period, compared with the period before and after the strike (Table 10). In Ashanti region, for instance, 2 deliveries were conducted on average during the strike, whereas the average number of deliveries before and after the strike stood at 5 and 15, respectively. In regions such as Bono and North East,

**Table 9. Percentage change in mean admissions during and after strike.**

| Region | Before Strike | During Strike | | After Strike | |
|---|---|---|---|---|---|
| | Mean Admissions | Mean Admissions | % Δ from Before Strike | Mean Admissions | % Δ from During Strike |
| Ashanti | 29 | 10 | -65.52 | 42 | +100 |
| Bono | 4 | 0 | -100 | 9 | +100 |
| Central | 20 | 12 | -40 | 7 | 41.67 |
| Greater Accra* | 13 | 0 | -100 | 4 | +100 |
| North East | 18 | 0 | -100 | 2 | +100 |

* = mean/average data

**Table 10. Percentage change in deliveries conducted during and after strike.**

| Region | Before Strike | During Strike | | After Strike | |
|---|---|---|---|---|---|
| | Mean Delivery | Mean Delivery | % Δ from Before Strike | Mean Delivery | % Δ from During Strike |
| Ashanti | 5 | 2 | -60 | 15 | +100 |
| Bono | 5 | 0 | -100 | 4 | +100 |
| Central | 4 | 3 | -25 | 8 | +100 |
| Greater Accra* | 1.5 | 0.5 | -66.67 | 0 | +100 |
| North East | 8 | 0 | -100 | 5 | +100 |
| Upper West | 3 | 2 | -33.33 | 4 | +100 |

* = mean/average data

there were no deliveries performed during the strike period. However, all the regions recorded more than 100% increase in deliveries conducted after the strike was called off.

### Reproductive health services

The result in Table 11 showed that, during the strike, no reproductive services were provided at Ashanti, Bono, Greater Accra and Upper West regions. Only Central region provided reproductive health services during the strike period. In Ashanti and Bono regions, more reproductive services were provided after the strike was called off, compared with before the strike was declared.

## Access to health services at health centers during strike

### Outpatient department attendance

As shown in Table 12, the average number of OPD attendance at health centers across the regions decreased during the strike period. Three of the regions (e.g., Greater Accra, Northern and Western regions) recorded no OPD attendance during the strike. For the remaining regions, the percentage decrease in average OPD attendance ranges from 46.86% (Western North region) to 94.49% (Upper West). Similarly, there was over 100% increase in the average OPD when the strike was called off.

### Deliveries conducted

Table 13 showed that the average deliveries conducted in the health centers reduced significantly during the strike period. The percentage decrease ranges from 14.29% (Western North) to 100% (e.g., Northern region). Following the suspension of the strike, the average deliveries

**Table 11. Percentage change in reproductive service use (RSU) during and after strike.**

| Region | Before Strike | During Strike | | After Strike | |
|---|---|---|---|---|---|
| | Mean RSU | Mean RSU | % Δ from Before Strike | Mean RSU | % Δ from During Strike |
| Ashanti | 256 | 0 | -100 | 251 | +100 |
| Bono | 8 | 0 | -100 | 12 | +100 |
| Central | 2 | 5 | +100 | 4 | -20 |
| Greater Accra* | 40 | 0 | -100 | 37 | +100 |
| Upper West | 3 | 0 | -100 | 2 | +100 |

* = mean/average data

**Table 12. Percentage change in the mean OPD attendance during and after strike.**

| Region | Before Strike | During Strike | | After Strike | |
|---|---|---|---|---|---|
| | Mean OPD Attendance | Mean OPD Attendance | % Δ from Before Strike | Mean OPD Attendance | % Δ from During Strike |
| Ashanti | 44.25 | 11.08 | -74.96 | 48.92 | +100 |
| Bono | 66 | 12 | -81.82 | 132 | +100 |
| Central | 53.62 | 5.92 | -88.96 | 55.23 | +100 |
| Greater Accra | 36 | 0 | -100 | 31 | +100 |
| North East | 105.33 | 33 | -68.67 | 111.33 | +100 |
| Northern | 16 | 0 | -100 | 12 | +100 |
| Savannah | 10.83 | 3 | -72.30 | 12.17 | +100 |
| Upper East | 40.5 | 2.75 | -93.21 | 37.75 | +100 |
| Upper West | 31.75 | 1.75 | -94.49 | 24.25 | +100 |
| Volta | 82 | 14.5 | -82.32 | 94.5 | +100 |
| Western | 18.67 | 0 | -100 | 26 | +100 |
| Western North | 28.6 | 15.2 | -46.85 | 36.2 | +100 |

**Table 13. Percentage change in the mean deliveries conducted during and after strike.**

| Region | Before Strike | During Strike | | After Strike | |
|---|---|---|---|---|---|
| | Mean Deliveries | Mean Deliveries | % Δ from Before Strike | Mean Deliveries | % Δ from During Strike |
| Ashanti | 2.75 | 0.58 | -78.91 | 2.58 | +100 |
| Bono | 0.5 | 0 | -100 | 1 | 100 |
| Central | 1.69 | 0.67 | -60.36 | 2.69 | +100 |
| Greater Accra | 1 | 0.5 | -50 | 4 | +100 |
| North East | 0.83 | 0.17 | -79.52 | 1 | 100 |
| Northern | 1 | 0 | -100 | 0 | 0 |
| Savannah | 1.17 | 1.5 | 28.21 | 1 | +100 |
| Upper East | 1.75 | 0.75 | -57.14 | 1.5 | +75 |
| Upper West | 0.5 | 0.38 | -24 | 0.75 | +100 |
| Volta | 2 | 0 | -100 | 3.5 | +100 |
| Western | 1.33 | 0 | -100 | 0.33 | +100 |
| Western North | 1.4 | 1.2 | -14.29 | 1.6 | +33 |

conducted increased beyond 100% from during the strike period for all but one region (Northern region) where there were no deliveries conducted.

## Discusion

Nurses and midwives are extremely important in the provision of healthcare services in Ghana, as stated previously. The health system is functional and operational due to the contributions of nurses and midwives. As a result, the strike action embarked on by the GRNMA from 21[st] septermber 2020 to 23[rd] September 2020 had significant impacts on access to health care services.

The results of the study showed that access to healthcare services decreased significantly across health facilities during the strike period. The average number of patients on admission before and after strike were more than twice the average number during the strike period. The reduction in healthcare deliveries was evident across the range of healthcare services investigated, including OPD attendance, admissions, deliveries, and surgeries. The findings reported

here largely support previous studies from other African countries, including Kenya [11], Nigeria [10] and Egypt [8] that showed a reduction in healthcare services utilization following strike actions by nurses and other health professionals.

A major strength of the current study is the wider focus. Unlike the narrowed focus of previous studies [6, 8, 9], this study focused on the impact of strike on a range of healthcare services from the 16 administrative regions of Ghana. Across the regions, there was a significant reduction in reproductive health service utilization. In three of the regions (i.e., Ahafo, Central and Northern), no one accessed reproductive health services during the strike period. The average number of people who utilized antenatal services decreased during the period of strike, compared with the period before and after the strike. In the North East and Northern regions, for example, there was zero use of antenatal services during the strike. Our study has shown that strike action by nurses and midwives has the propensity to negatively impact access to a range of healthcare services provided by different units or departments in different administrative or geographical locations.

The decrease in healthcare access could be attributed to the fact that nurses and midwives are the "basic unit" of the healthcare system such that in their absence, the public is not able to initiate help-seeking behavior. Indeed, nurses activate the healthcare system through the provision of basic health services such as temperature, blood pressure and blood glucose that form the foundation for advanced and detailed healthcare services. Besides, nurses provide major support to other health professionals in rendering healthcare services. Surgeries, for instance, cannot be performed by only surgeons without the professional skills of theatre nurses. Surgical cases require admissions before and after the surgical procedure. Nurses generally manage the admission processes. Without adequate number of nurses to care for patients before and after surgery, undesirable surgical outcomes, including complication and wound infection, may manifest quite easily and quickly. Therefore, the absence of nurses tends to create a significant professional gap that other health professionals, by virtue of their training and specialization, are not able to immediately and appropriately fulfill their roles. Because the public is aware of the duties and responsibilities nurses and midwives play in the health system, news about their strike could affect help-seeking decision from healthcare facilities.

The reduction in access to healthcare services have enormous consequences, as demonstrated by previous studies. For example, strike action by healthcare workers has been linked to increased mortalities [13]. The absence or limited supply of vaccination services occasioned by strike action [11] could complicate efforts to improve the health of children and adolescent as well as preventing death from childhood diseases. Although data were not obtained on service users; however, previous studies have shown that social and economically disadvantaged groups are extremely vulnerable to the negative repercussions of strike as they are unable to pay for services in a private hospital [19]. These individuals may resort to the use of herbal preparations and self-medication that often exacerbate their conditions, culminating into increased morbidity and mortality [9, 10, 19].

A major caveat in the literature is the limited data on access to healthcare services following the suspension of strike action by healthcare professionals. To fill this void, the current study compared healthcare services utilization during and after the strike action by nurses and midwives. As reported previously, service used increased drastically following the cessation of the strike action. The increase in access to healthcare services is a testament of the unmet health needs of the public. The decrease in the provision of healthcare services during the strike period could drive the public towards private healthcare system. However, most Ghanaians cannot afford the services of private healthcare facilities owing to the relatively high cost of treatment [20]. These individuals are left with the option of waiting on the news of resumption of work by nurses and midwives so that they can access healthcare. The drastic increase in the

use of healthcare services could be due to several factors, including the fear of another strike. Indeed, the strike was called off following an injunction order secured from a court of competent jurisdiction by the National Labor Commission (NLC) of Ghana, who questioned the legitimacy of the strike. The NLC, which is mandated to address labor agitation and unrest issues in Ghana, does not negotiate with striking labor unions. Like in other jurisdiction, the labor laws favor strike. However, the processes to do so have always been so onerous and time-consuming that unions in Ghana rarely engage in legally recognized strikes [21]. The public could be worried that the negotiation between the NLC and GRNMA may fail, thereby providing the motivation for another strike action. Indeed, instances of failed negotiation between the NLC and labor unions are very common development in Ghana. The public, therefore, saw the immediate resumption of duty as the best opportunity to obtain healthcare services before another strike is announced. There is also the possibility that people visited the health facilities to learn for themselves whether the healthcare delivery process has normalized, following its truncation by the strike actions. In the process, they might have accessed healthcare services, adding to the number of service users after the strike period.

## Limitations of the study

The study findings should be evaluated considering the following limitations. The study was designed to gather data from the health facilities located across the 16 administrative regions of Ghana. However, as a typical nature of research, we could not obtain data from all the facilities across the regions or districts. Some heads of facilities did not grant approval for the data gathering process. In some instances, we could not identify individuals interested in supporting the research as field officers. The data reported here were based on facilities that were somewhat conveniently selected. The lack of random selection of the health facilities limits the generalization of the findings reported in this study. Given the nature of the research design, particularly the data collection method and data collected, it was impossible to determine whether and to what extent other factors beyond the strike action contributed to the decrease in access to healthcare during strike period and an increase post-strike.

## Conclusions

The strike action by the nurses and midwives in Ghana significantly reduced access to a range of health services, including reproductive health services, deliveries, and antenatal services. The distribution of the impacts of the strike action across the regions is largely uniform. Likewise, the impact of the strike action extended to various types of health facilities, hospitals, polyclinics and health centers. The far-reaching impact suggest clearly that strike action by nurses and midwives will derail the efforts and investment made towards achieving the universal healthcare coverage by the Government of Ghana and development partners. It is also imperative to state that the attainment of the goal 3 (health and well-being) of SDG in Ghana will be negatively affected with strike action by nurses and midwives. This understanding calls for proactive measures by the Government of Ghana and relevant organizations to respond adequately and appropriately to the needs and concerns of nurses and midwives as the largest healthcare workforce. Doing so requires some practical steps such as periodic assessment of the wellbeing and conditions of service of nurses and midwives and provide the necessary support system that accommodate existing socioeconomic conditions. Increasing cost of goods and services, for example, should be accompanied by a corresponding increase in salary or financial incentives. This basic economic principle would mitigate against the financial challenges suffered by nurses and midwives, significantly reducing agitation for improved

conditions of services amidst economic hardship induced by the raising inflation and cost of petroleum products.

## Supporting information

**S1 File. Raw data used for the analysis.**
(XLS)

## Author Contributions

**Conceptualization:** Perpetual Ofori Ampofo, David Tenkorang-Twum, Samuel Adjorlolo, Margaretta Gloria Chandi, Francis Kwaku Wuni, Vida Ami Kukula, Sampson Opoku.

**Data curation:** David Tenkorang-Twum, Samuel Adjorlolo, Margaretta Gloria Chandi, Francis Kwaku Wuni, Ernestina Asiedu.

**Investigation:** Francis Kwaku Wuni.

**Methodology:** Samuel Adjorlolo, Margaretta Gloria Chandi, Francis Kwaku Wuni, Ernestina Asiedu, Vida Ami Kukula.

**Project administration:** Perpetual Ofori Ampofo, David Tenkorang-Twum, Samuel Adjorlolo, Vida Ami Kukula.

**Resources:** Perpetual Ofori Ampofo, David Tenkorang-Twum, Ernestina Asiedu.

**Supervision:** Perpetual Ofori Ampofo.

**Writing – original draft:** Samuel Adjorlolo, Francis Kwaku Wuni, Ernestina Asiedu, Vida Ami Kukula.

**Writing – review & editing:** Samuel Adjorlolo, Francis Kwaku Wuni, Ernestina Asiedu, Vida Ami Kukula, Sampson Opoku.

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
