## [Decision Letter · Decision Letter 0]

31 May 2022

PONE-D-21-35187Macro-level Impact of Strike Action by Ghana Registered Nurses and Midwives.PLOS ONE

Dear Dr. Adjorlolo,

Thank you for submitting your manuscript to PLOS ONE. After careful consideration, we feel that it has merit but does not fully meet PLOS ONE’s publication criteria as it currently stands. Therefore, we invite you to submit a revised version of the manuscript that addresses the points raised during the review process.

Please note that we have only been able to secure a single reviewer to assess your manuscript. We are issuing a decision on your manuscript at this point to prevent further delays in the evaluation of your manuscript. Please be aware that the editor who handles your revised manuscript might find it necessary to invite additional reviewers to assess this work once the revised manuscript is submitted. However, we will aim to proceed on the basis of this single review if possible. 

The reviewer has raised a number of concerns that need attention. They request additional information on methodological aspects of the study and they question the internal and external validity of the results reported.

We look forward to receiving your revised manuscript.

Kind regards,

Thomas Phillips, PhD

Staff Editor

PLOS ONE

**Journal requirements:**

“Not applicable”

“None has been declared”

4. Please upload a copy of Figure 4,7 and 9, to which you refer in your manuscript. If the figure is no longer to be included as part of the submission please remove all reference to it within the text.

Reviewers' comments:

Reviewer's Responses to Questions

**Comments to the Author**

1. Is the manuscript technically sound, and do the data support the conclusions?

Reviewer #1: Yes

2. Has the statistical analysis been performed appropriately and rigorously? 

Reviewer #1: N/A

3. Have the authors made all data underlying the findings in their manuscript fully available?

Reviewer #1: Yes

4. Is the manuscript presented in an intelligible fashion and written in standard English?

Reviewer #1: Yes

5. Review Comments to the Author

Reviewer #1: Overall

-Interesting study that shows the lack of access to care for the public when healthcare workers go on strike. See below a few suggestions, and questions for clarification to improve how the story is presented.

Title

-Suggest, rewording the title, as it does not only focus on the macro level of the health system (which I understand to be the national level). The study focuses on hospitals and health centres across different regions suggesting a multi-level approach.

Abstract

-Suggest similar wording in the abstract regarding how many facilities were accessed for data collection as in the methods section (see comment on article)

Introduction and Background

-The introduction needs improvement to have better flow. Currently the integration mixes information about HCW strikes from various strikes across different African countries, thus it is not quite clear what gap this specific paper is filling in the already existing literature, or what exactly it will be adding on. I suggest re-ordering of the information presented starting with what is known about HCW strikes worldwide and their effects, what is known in Africa, and then in Ghana, then stating which areas the literature (especially from Africa and Ghana) has paid less attention to, and then stating as has been well-stated in the paper what gaps the paper will be filling.

-Would also be useful to tell the reader a bit more about the Ghana health system context. This is included in the project setting, but suggest moving to the introduction

-Suggest carefully re-reading of the introduction to ensure adequate referencing of studies e.g. the sentence on the longest ever strike action needs referencing. Other statements about studies done elsewhere reporting on low immunization rates during HCW strikes need citation too (pgs 4-6).

-Also re-read to check typos and sentence structure to improve clarity. See comments on the paper

Methods

-Project setting section states data was collected from 155 government-owned facilities while results report on data from 188 facilities. What of the other remaining facilities, who owned them?

-Not clear what/which hospitals or health centres were included for every region. Was it every hospital and/or health centre in the region. If not, what was the selection criteria for the health facilities selected for data collection?

-Suggest moving the information on the Ghana health system (first three sentences could be moved to the introduction part)

-Some of the information on the section on training field officers is repetitive. Suggest summarising in one or two sentences and re-wording for simplicity, for example by mentioning what the field officers were trained on, and on which platform training occurred. Overall suggest reduction of information about the field officers (e.g. information on how much they were paid etc not necessary)

-Could you clarify why you chose to use of field officers who were nurses to collect data on service delivery (or its absence) yet they were the ones on strike?

-Regarding your variables, aren’t reproductive health services inclusive of ANC, and delivery services? Did reproductive health services only refer to family planning services? If this is the case, then I suggest to use the term family planning services instead for clarity.

-Curious why immunisation services were not included as among the variables reported in the study? In most Low and Middle Income Countries (LMICs) these are provided by nurses and have been shown to be affected when healthcare workers’ strikes occur.

Results

-Suggest a brief introduction that summarises what results will be presented. This could also include a sentence reminding the reader that the data on hospitals will be presented separately.

Discussion

-Discussion refers to decline in child health services, yet these are not reported on in the results section. Please clarify which child health services were affected…are these immunisation services? Or consultation for children with illnesses in the OPD? If the latter, important to make this distinction in the results section

-The discussion also refers to the decline in surgical services during the strike and links this to mortalities, however no mortality data has been reported. The data does not show us a link between the fewer surgeries conducted and mortality, either from this study or other published literature. It’s necessary to justify this claim, especially because a few studies have shown the absence of a significant increase in mortality during healthcare worker strike periods (see Ong'ayo, Gerald, et al. "Effect of strikes by health workers on mortality between 2010 and 2016 in Kilifi, Kenya: a population-based cohort analysis." The Lancet Global Health 7.7 (2019): e961-e967; Cunningham, Solveig Argeseanu, et al. "Doctors' strikes and mortality: a review." Social science & medicine 67.11 (2008): 1784-1788.)

-One of the study limitations was the lack of random selection of facilities for inclusion in this study. However, it has not been clarified, here or in the methods section, how the facilities were selected

-The discussion hardly interprets the study findings in the context of other existing literature on healthcare worker strikes, either in Ghana or other parts of the world. I suggest that some of the papers referred to in the introduction could be revisited in the discussion to compare their findings and this study’s findings including what new angle, the study adds to our understanding of healthcare worker strikes and their effects. If there are differences between existing study and other literature, then useful to suggest reasons why this might be the case. The introduction could then be more focused on explaining for example the context of Ghana, previous histories of strikes, and why it is important to address the issue of HCW strikes.

-Another point of discussion might be if for services provided by both hospitals and health centres (e.g. ANC services, deliveries) which were more affected, and why this might have been the case. This would require perhaps linking back to the set up of the Ghana health system.

6. PLOS authors have the option to publish the peer review history of their article (what does this mean?). If published, this will include your full peer review and any attached files.

Reviewer #1: No

---

## [Author Response · Author response to Decision Letter 0]

30 Jul 2022

Find attached the document named "Response to Reviewer Comments"

---

## [Decision Letter · Decision Letter 1]

4 Sep 2022

PONE-D-21-35187R1A Multi-Level Impact Analysis of Strike Action by Ghana Registered Nurses and Midwives on Access to and Utilization of Healthcare Services.PLOS ONE

Dear Dr. Adjorlolo,

Thank you for submitting your manuscript to PLOS ONE. After careful consideration, we feel that it has merit but does not fully meet PLOS ONE’s publication criteria as it currently stands. Therefore, we invite you to submit a revised version of the manuscript that addresses the points raised during the review process.

ACADEMIC EDITOR: 

Dear Authors, 

1.We invite you to make further reviews based on the comments of the reviewer as attached. 

2. We noticed that your manuscript did not have ethical approval possibly because it was a retrospective anonymous data analysis. However, we suggest that you obtain a communication from your local ethics review board that the research do not require an ethical approval and that the dissemination of the research may not violate any ethical principles. You may then adjust your ethics statement to include such communication from ethics committee.

3. Title: "Multi-level ...." is usually reserved for a specific statistical analysis which was not conducted in this study. It is suggested that the title should be revised so as not to confuse the audience

4. First objective: Please change "utilization" to "utilizing"

5. For ease of review, Please provide a continuos number line while revising your manuscript

6. Results: Please delete "In this section, we presented the results of the data analyses, commencing with the distribution of healthcare facilities. This was followed by the presentation of findings relating to the utilization of healthcare services. This was structured based on the type of health facilities, starting first with hospitals, followed by polyclinics, and lastly health centers. We examined the extent of and change in use of the various health services described under study variables for the various health facility types. This was done separately for each region"

7. References: The references are not in line with the referencing style of PLOS ONE. (See: Submission Guidelines | PLOS ONE)

We look forward to receiving your revised manuscript.

Kind regards,

Gbenga Olorunfemi, MBBS,MSC,FMCOG,FWASC

Academic Editor

PLOS ONE

Reviewers' comments:

Reviewer's Responses to Questions

**Comments to the Author**

1. If the authors have adequately addressed your comments raised in a previous round of review and you feel that this manuscript is now acceptable for publication, you may indicate that here to bypass the “Comments to the Author” section, enter your conflict of interest statement in the “Confidential to Editor” section, and submit your "Accept" recommendation.

Reviewer #1: All comments have been addressed

2. Is the manuscript technically sound, and do the data support the conclusions?

Reviewer #1: Yes

3. Has the statistical analysis been performed appropriately and rigorously? 

Reviewer #1: Yes

4. Have the authors made all data underlying the findings in their manuscript fully available?

Reviewer #1: Yes

5. Is the manuscript presented in an intelligible fashion and written in standard English?

Reviewer #1: Yes

6. Review Comments to the Author

Reviewer #1: Well-written article addressing an important health system issue. The problem being addressed is well described in the introduction, and the study objectives are clear. Findings are presented clearly and link well with the discussion and study conclusions. A few minor typos identified in the text. These are highlighted in the attached document.

7. PLOS authors have the option to publish the peer review history of their article (what does this mean?). If published, this will include your full peer review and any attached files.

Reviewer #1: No

---

## [Author Response · Author response to Decision Letter 1]

20 Sep 2022

Comment 1: We noticed that your manuscript did not have ethical approval possibly because it was a retrospective anonymous data analysis. However, we suggest that you obtain a communication from your local ethics review board that the research do not require an ethical approval and that the dissemination of the research may not violate any ethical principles. You may then adjust your ethics statement to include such communication from ethics committee.

Authors’ response: Thank you very much for the recommendation. We have received a response from the local ethics and have included same in the ethics section of the manuscript.

Comment 2: Title: "Multi-level ...." is usually reserved for a specific statistical analysis which was not conducted in this study. It is suggested that the title should be revised so as not to confuse the audience.

Authors response: The title has been modified to “The Impact of Strike Action by Ghana Registered Nurses and Midwives on the Access to and Utilization of Healthcare Services” (Page 1, line 1-2, page 2, lines 27)

Comment 3: First objective: Please change "utilization" to "utilizing"

Authors response: The change has been applied. (Page 6, line 140).

Comment 4: For ease of review, Please provide a continuous number line while revising your manuscript.

Authors response: This has been provided.

Comment 5: Results: Please delete "In this section, we presented the results of the data analyses, commencing with the distribution of healthcare facilities. This was followed by the presentation of findings relating to the utilization of healthcare services. This was structured based on the type of health facilities, starting first with hospitals, followed by polyclinics, and lastly health centers. We examined the extent of and change in use of the various health services described under study variables for the various health facility types. This was done separately for each region"

Authors response: This section has been deleted.

Comment 6: References: The references are not in line with the referencing style of PLOS ONE. (See: Submission Guidelines | PLOS ONE)

Authors response: The reference has been formatted in line with the style of PLOS ONE.

---

## [Editor Report · Decision Letter 2]

21 Sep 2022

The Impact of Strike Action by Ghana Registered Nurses and Midwives on the Access to and Utilization of Healthcare Services

PONE-D-21-35187R2

Dear Dr. Adjorlolo,

We’re pleased to inform you that your manuscript has been judged scientifically suitable for publication and will be formally accepted for publication once it meets all outstanding technical requirements.

Kind regards,

Gbenga Olorunfemi, MBBS,MSC,FMCOG,FWASC

Academic Editor

PLOS ONE

Additional Editor Comments (optional):

Dear Authors,

Please may you attend to this minor corrections

1. Line 224 - 225: Delete "Given the nature of the study, we did not seek IRB approval"

2. Citations: Please be consistent with the citations. You used "square" reference and at some other point you used "circular" reference. See line 85 and line 86 , line 90 and line 95 for example
---

## [Editor Report · Acceptance letter]

3 Oct 2022

PONE-D-21-35187R2 

The Impact of Strike Action by Ghana Registered Nurses and Midwives on the Access to and Utilization of Healthcare Services 

Dear Dr. Adjorlolo:

I'm pleased to inform you that your manuscript has been deemed suitable for publication in PLOS ONE. Congratulations! Your manuscript is now with our production department. 

Kind regards, 

on behalf of

Dr. Gbenga Olorunfemi 

Academic Editor

PLOS ONE